# Predictive accuracy of risk prediction models for recurrence, metastasis and survival for early-stage cutaneous melanoma: a systematic review

Tafadzwa Patience Kunonga [1,2] R P W Kenny,[1,2] Margaret Astin,[1]
Andrew Bryant,[3] Vasileios Kontogiannis,[4] Diarmuid Coughlan,[4]
Catherine Richmond,[1,2] Claire H Eastaugh,[1,2] Fiona R Beyer,[1,2] Fiona Pearson,[1,2]
Dawn Craig,[1,2,4] Penny Lovat,[5,6] Luke Vale,[4] Robert Ellis[5,6,7]

For numbered affiliations see end of article.

**Correspondence to**
Tafadzwa Patience Kunonga;
patience.kunonga@newcastle.ac.uk

## ABSTRACT

**Objectives** To identify prognostic models for melanoma survival, recurrence and metastasis among American Joint Committee on Cancer stage I and II patients postsurgery; and evaluate model performance, including overall survival (OS) prediction.

**Design** Systematic review and narrative synthesis.

**Data sources** Searched MEDLINE, Embase, CINAHL, Cochrane Library, Science Citation Index and grey literature sources including cancer and guideline websites from 2000 to September 2021.

**Eligibility criteria** Included studies on risk prediction models for stage I and II melanoma in adults ≥18 years. Outcomes included OS, recurrence, metastases and model performance. No language or country of publication restrictions were applied.

**Data extraction and synthesis** Two pairs of reviewers independently screened studies, extracted data and assessed the risk of bias using the CHecklist for critical Appraisal and data extraction for systematic Reviews of prediction Modelling Studies checklist and the Prediction study Risk of Bias Assessment Tool. Heterogeneous predictors prevented statistical synthesis.

**Results** From 28 967 records, 15 studies reporting 20 models were included; 8 (stage I), 2 (stage II), 7 (stages I–II) and 7 (stages not reported), but were clearly applicable to early stages. Clinicopathological predictors per model ranged from 3–10. The most common were: ulceration, Breslow thickness/depth, sociodemographic status and site. Where reported, discriminatory values were ≥0.7. Calibration measures showed good matches between predicted and observed rates. None of the studies assessed clinical usefulness of the models. Risk of bias was high in eight models, unclear in nine and low in three. Seven models were internally and externally cross-validated, six models were externally validated and eight models were internally validated.

**Conclusions** All models are effective in their predictive performance, however the low quality of the evidence raises concern as to whether current follow-up recommendations following surgical treatment is

## STRENGTHS AND LIMITATIONS OF THIS STUDY

⇒ Comprehensive and systematic searches captured the core evidence about prediction models in early-stage melanoma.
⇒ The current evidence reviewed across all models is insufficient to make recommendations regarding use in clinical practice.
⇒ Heterogeneity in the data across study populations, predictors and clinical progression of the disease suggests there is insufficient evidence to make firm conclusions about best clinical practice in the application of predictive models in patients with American Joint Committee on Cancer stage I or II cutaneous melanoma.

adequate. Future models should incorporate biomarkers for improved accuracy.

**PROSPERO registration number** CRD42018086784.

## INTRODUCTION

Melanoma is one of the deadliest of all skin cancers.[1] The worldwide incidence of melanoma is estimated to be approximately 2% of cancers per annum.[2] However, in early-stage disease, once surgically removed from the skin, through wide local excision (WLE), and without adjuvant immunotherapy, chemotherapy or radiotherapy, the majority of early-stage melanomas are cured, with an estimated 5-year overall survival (OS) rate of 95%–100% (stage I) and 65%–92.8% (stage II).[3] However, up to 30% of all primary melanomas progress to metastatic disease with an associated extremely poor survival rate of only 5%–15%.[4] Although the surgical treatment of primary melanoma is effective and long established, there has been a rapid pace of change recently with the addition of sentinel lymph node biopsy (SLNB).[5] This

procedure identifies the sentinel lymph node which is excised for examination to determine whether cancer cells are present.[6]

Currently, a structured, uniformly adopted, evidence-based model of patient follow-up after initial diagnosis and treatment is lacking.[7] Follow-up strategies depend on the stage and progression of the disease; based on the criteria defined within the eighth edition of the American Joint Committee on Cancer (AJCC) staging manual.[8] For this review, however, the definitions of stage and progression used in the 2008, seventh edition[9] and earlier editions remain relevant as the reported data in research studies may be based on definitions from earlier editions. Current guidelines for management of the condition vary across the world, with most developed using anecdotal evidence and expert opinion. A clinical prediction model, a statistical tool that relates multiple factors to the probability of a patient having a future clinical event,[10] could be used to aid early detection of disease progression such as new primary tumours, in-transit metastasis or locoregional recurrence in the regional lymph nodes. Before introducing a new clinical prediction model into practice there needs to be evidence of model development, validation and impact. Validation should involve evaluating performance (in terms of discrimination or calibration), and clinical usefulness in practice. Impact should consider whether the model improves decision-making.[10]

Previous systematic reviews have been conducted looking at the value of risk prediction models for melanoma development,[11–13] and clinical risk of survival of melanoma.[14] However, to our knowledge, no review has critically appraised current clinicopathological prediction models for primary melanomas (stages I and II), following surgical excision of the tumour. This review aims to identify and assess prognostic/predictive models used to predict patient survival, recurrence (any site) and metastasis in AJCC stage (I and II) melanoma following surgical excision, thus potentially allowing further refinement of risk-stratification of patients.

## METHODS
The review population listed in the protocol was later augmented to include stage II melanoma. As with stage I melanoma, tumours at this stage may not have advanced and are treated with WLE.[15] The review followed the Preferred Reporting Items for Systematic Reviews and Meta-Analyses guideline (PRISMA).[16]

### Search strategy
Searches were originally conducted in July 2019, then updated in September 2021. A date restriction of 2000 was used, as this is when SLNB became routine in the USA.[5] The search strategy (online supplemental tables 1 and 2) was designed in MEDLINE and translated to other databases (online supplemental table 3). A published and validated prognostic study filter[17] was used including the following concepts: (melanoma) AND (risk models) AND (prognosis). No language or country of publication restrictions were applied. In the search update (June 2019 to September 2021), a more up to date search filter was used.[18]

### Inclusion and exclusion criteria
Studies were selected using piloted inclusion criteria (table 1) according to the Patient population, Intervention, Comparator, Outcome, Timing and Setting formula.[19] The primary outcome is OS defined as patient survival until death from melanoma following primary treatment. Secondary outcomes include: number of detected recurrence and metastasis, and the discrimination, calibration, overall performance, and clinical utility of the models. A table of definitions for the performance measures is presented in Online supplemental table 4.

### Selection of studies
Two reviewers independently screened titles and abstracts for articles relevant to stage I disease. For stage II screening a text mining approach was adopted, followed by duplicate and independent hand screening of study abstracts. Two reviewers independently screened the selected full text articles. For each included study, data were extracted by one of four reviewers using the CHecklist for critical Appraisal and data extraction for systematic Reviews of prediction Modelling Studies (CHARMS) checklist.[20] Data were extracted for characteristics of participants, outcomes, predictors, model development methods, model performance and validation. Extraction forms were checked for accuracy and consistency by another reviewer, and a clinical expert.

### Risk of bias assessment
Two pairs of reviewers used the Prediction study Risk of Bias Assessment Tool (PROBAST),[21] to assess the risk of bias (ROB) of each study. The tool, which has gained popularity on reviews of prognostic studies,[22] evaluates 20 questions in four domains (Participants, Predictors, Outcome and Analysis) to assess the ROB and applicability of prediction model studies.[21] Following the guide, we used the signalling questions across these domains to make the following judgements: 'yes' means low bias, 'no' means high bias. For the overall judgement, the study was rated as low ROB if all domains were rated as low, high ROB if at least one domain was rated as high. Where there was insufficient information to make a judgement, we rated the domain as unclear ROB. However, if one domain was rated as unclear and the others low, overall judgement was rated as unclear.[21]

### Data analysis and synthesis
A narrative synthesis was undertaken, including summarising the characteristics of the included models and the performance of prediction models, focusing on measures of discrimination, calibration and overall performance; and model validation methods. We were unable to conduct a meta-analysis due to the variance in use of disparate predictors, and modelling methods.

**Table 1**  Inclusion criteria

| Criteria | Include | Exclude |
|---|---|---|
| Population | Adults aged ≥18 years treated for American Joint Committee on Cancer (seventh edition) stage I and II cutaneous melanoma<br>▶ Stage IA (T1a ≤1 mm thick); or<br>▶ Stage IB (T1b with ulceration or mitoses ≤1 mm thick, or T2a 1.01 to 2.00 mm thick and no ulceration); or<br>▶ Stage IIA (1.01–2.00 mm thick with ulceration, or 2.01–4.00 mm thick without ulceration); or<br>▶ stage IIB (2.01–4.00 mm thick with ulceration, or >4.0 mm without ulceration)<br>▶ stage IIC (>4.0 mm with ulceration) | Advanced melanoma<br>Metastatic melanoma<br>Stage III<br>Stage IV |
| Types of prognostic models | ▶ Nomogram<br>▶ Scoring system<br>▶ Equation<br>▶ Classification or decision trees<br>*Minimum of two clinicopathological factors | Models built with predictive biomarker factors included<br>Models built with gene expression profiling<br>Diagnostic studies<br>Studies assessing only one predictor |
| Outcome measures | Primary: overall survival<br>Secondary: recurrence (any site), metastases, prognostic performance of risk models | |
| Timing | Postresection of the primary cutaneous tumour<br>▶ Diagnostic excision<br>▶ Re-excision, also called wide local excision (WLE) | Studies that are looking at treatment (eg, therapy) of melanoma |
| Setting | Any setting (primary, secondary or tertiary care) | |
| Study design | Retrospective cohort studies<br>Prospective cohort studies<br>Randomised controlled trials | Non-empirical studies |

Furthermore, it was not possible to perform subgroup analyses as the studies identified did not report outcomes by population subgroups.

### Quality of the evidence

We had planned to assess the overall quality of evidence using the Grading of Recommendations Assessment, Development and Evaluation.[23] However, although the tool has been adapted for assessing overall quality in prognostic factor reviews, studies, it is yet to be adapted for prediction model reviews.[24]

### Patient and public involvement

This review was requested by our funder within a timescale that did not allow for meaningful public and patient involvement.

### RESULTS

### Selection of studies

We identified 28 967 records after deduplication, of which 165 were selected for full text screening (online supplemental figure 1). Fourteen studies reporting twenty unique risk prediction models, met the inclusion criteria.[25–38]

### Characteristics of studies

Characteristics of included studies are presented in table 2. Studies were published between 2000 and 2021; nine from the USA,[26 27 30 31 33–35 37 38] two from Australia,[25 28] two from Europe[32 36] and one from Brazil.[29] Twelve studies were from retrospective cohorts[25–32 36–38] and three were from prospective cohorts.[33–35] Patient data were obtained from population-based cancer registries, medical/clinical records or a combination of these sources. Study intervals ranged from 1972 to 2015. Eight models were developed from patients with stage I melanoma only (three from one study,[28] two from another study[30] and three from three studies[31–33]), two for stage II only (both reported in the same study[29]), eight for stages I–II (two reported in one study[36] and five from five studies[25 26 34 35 38]) and for three models the stages not reported (two from one study[27] and the other from one study[37]), however they were found to be applicable to stages I–II. Eleven studies used logistic regression methods to develop the models,[25–29 32–34 36–38] and three used a recursive partitioning method.[30 31 35] Duration of follow-up ranged from 3[38] to 20 years.[26]

### Characteristics of included models

Characteristics of included models are presented in table 3. Model outcomes included OS[25 27 30 32 34 35 38]; melanoma specific survival (MSS)[25 26 29]; melanoma specific

**Table 2** Characteristics of included studies

| Citation | Study location | Study design | Data source (centres (n)) | AJCC disease stage/ clinical staging | Study interval | Development method |
|---|---|---|---|---|---|---|
| Baade et al[25] | Australia | Retrospective | Population based Queensland cancer registry | 2002 AJCC 94% Stages I–II | 1995–2008 | Multivariate analysis |
| Balch et al[26] | USA | Retrospective | Population based AJCC database (13) | Stages I–II | Not reported | Multivariate analysis |
| Cochran et al[27] | USA | Retrospective | John Wayne Cancer Institute Melanoma clinical database, Division of Surgical Oncology, UCLA | 1988 AJCC Not reported | 1980–1990 | Multivariate analysis |
| El Sharouni et al[28] | Netherlands Australia | Retrospective | DEV: PALGA, the Dutch Pathology Registry VALID: Melanoma Institute Australia | 2009 AJCC Stages IA–IB IA (T1a)=58.6% IB (T1b)=29% T1nos=12.4% | 2000–2014 | Multivariate analysis |
| Fonseca et al[29] | Brazil | Retrospective | C. Camargo Cancer Centre database | 2009 AJCC Stages IIB–IIC | 2000–2014 | Multivariate analysis |
| Gimotty et al[31] | USA | Retrospective | DEV: SEER population based registry VALID: New SEER patients | 2002 AJCC Stages IA–IB IA (T1a) = 86% IB (T1b) = 14% | DEV: 1972–1991 VALID: 1991–1995 | Recursive partitioning |
| Gimotty et al[30] | USA | Retrospective | DEV: SEER Registry VALID: Clinical based PLC registry | 2002 AJCC Stages IA (T1a)–IB (T1b) | DEV: 1998–2001 VALID: 1972–2001 | Recursive partitioning |
| Maurichi et al[32] | Europe | Retrospective | European clinical based centres (6) | 2009 AJCC Stages IA–IB T1a=50.3% T1b=49.7% | 1996–2004 | Multivariate analysis |
| Rosenbaum et al[33] | USA | Prospective | NYU clinicopathological biospecimen database | 2009 AJCC Stage IB | 2002–2014 | Multivariate analysis |
| Soong et al[34] | USA | Prospective | DEV: 2008 AJCC population based Melanoma Database (9) VALID: Sydney Melanoma Unit, Australia | 2008 AJCC Stages I–II | DEV: 26% diagnosed after 2002 VALID: Not reported | Multivariate analysis |
| Tsai et al[35] | USA | Prospective | AJCC Melanoma population-based database (13) | 2002 AJCC Stages IA–IIC | Not reported | Recursive partitioning |
| Verver et al[36] | Europe | Retrospective | EORTC | 2009 AJCC Stages IA–IIC | 1997–2013 | Multivariate analysis |
| Vollmer and Seigler[37] | USA | Retrospective | University Melanoma Clinic database | 1988 AJCC Not reported | 1980–1990 | Multivariate analysis |
| Xiao et al[38] | USA | Retrospective | SEER Registry | 2009 AJCC Stages IA–IIC | 2010–2015 | Multivariate analysis |

AJCC, American Joint Committee on Cancer; DEV, development set; EORTC, European Organisation for Research and Treatment of Cancer Centres; NYU, New York University; RND, regional node dissection; SEER, Surveillance, Epidemiology and End Results Programme; UCLA, University of California, Los Angeles; VALID, validation set.

mortality (MSM)[36]; mortality[37]; recurrence[27 36]; recurrence free survival (RFS)[33]; local RFS (LRFS)[28]; regional RFS (RRFS)[28]; distant RFS (DRFS)[28 29] and metastasis.[31] A table of definitions used in the studies for model outcomes is presented in Online supplemental table 5. Total sample sizes across the models ranged from 259[29] to 66 192 patients.[38] Five studies selected candidate clinicopathological variables based on previous clinical knowledge or literature (pre-specification),[25 28 30 35 37] three conducted a multivariate analysis,[30 32 34] two studies conducted a univariate analysis,[29 33] while three employed a combination of previous knowledge and univariate analysis.[27 36 38] One study did not provide information on variable selection.[26] A backward elimination analysis approach was used in seven studies,[25 27–29 31 33 36] and a full model approach was used in six studies,[30 32 34 35 37 38] to select the final predictors for model development. The number of predictors in the final models ranged from 3[33 36] to 10.[38] The most common predictors were ulceration (17/20), Breslow thickness/depth (16/20), sociodemographic status (15/20) and body/anatomic site (14/20) (see figure 1). The models are reported based on

**Table 3** Model development

| Citation | Sample size | Follow-up period | End point | Events (n) | Candidate selection method | Final predictor selection | Final predictors (n) | Validation method |
|---|---|---|---|---|---|---|---|---|
| Baade et al[25] | 28 654 | Median=7.2 years (86.4 months) | MSS | 1700 | Prespecification from existing data | Backward elimination | 7 | Internal-external cross validation |
| Balch et al[26] | 13 581 | 5–20 years | MSS | NR | Unclear | Unclear | 8 | External—geographic |
| Cochran et al[27] | 1042 | Median=42.5 months | OS | NR | Prespecification+univariate analysis | Backward elimination | 5 | Internal sample validation |
| Cochran et al[27] | 1042 | Median=42.5 months | Recurrence | NR | Prespecification+univariate analysis | Backward elimination | 4 | Internal sample validation |
| El Sharouni et al[28] | DEV: 25 930 VALID: 2968 | DEV: Median=6.7 years VALID: Median=12 years | LRFS | 232 | Prespecification from existing data | Backward elimination | 6 | Internal-external cross validation |
| El Sharouni et al[28] | DEV: 25 930 VALID: 2968 | DEV: Median=6.7 years VALID: Median=12 years | RRFS | 564 | Prespecification from existing data | Backward elimination | 7 | Internal-external cross validation |
| El Sharouni et al[28] | DEV: 25 930 VALID: 2968 | DEV: Median=6.7 years VALID: Median=12 years | DRFS | 278 | Prespecification from existing data | Backward elimination | 6 | Internal-external cross validation |
| Fonseca et al[29] | 259 | Median=80.13 months | DRFS | 117 | Univariate analysis | Backward elimination | 5 | Bootstrap |
| Fonseca et al[29] | 259 | Median=80.13 months | MSS | 117 | Univariate analysis | Backward elimination | 5 | Bootstrap |
| Gimotty et al[31] | DEV: 884 VALID: 114 | At least 10 years | Metastasis | 127 | Multivariate analysis | Backward elimination | 4 | External—new cohort |
| Gimotty et al[30] | DEV: 26 114 | Median=4.6 years | OS | 3593 | Prespecification from existing data | Full model approach | 4 | External—geographic |
| Gimotty et al[30] | VALID: 2389 | Median=8.1 years | OS | 1076 | Prespecification from existing data | Full model approach | 5 | External—geographic |
| Maurichi et al[32] | 2243 | 124 months | 12 year OS | 240 | Multivariate analysis | Full model approach | 6 | Congruence examination |
| Rosenbaum et al[33] | DEV: 506 VALID: 149 | Median=4.4 years | RFS | NR | Univariate analysis | Backward elimination | 3 | Internal k-fold validation |
| Soong et al[34] | DEV: 14 760 VALID: 10 974 | 5 and 10 years | OS | NR | Multivariate analysis | Full model approach | 6 | External—geographic |
| Tsai et al[35] | 13 268 | Not reported | OS | NR | Prespecification from existing data | Full model approach | 6 | Internal-external cross validation |
| Verver et al[36] | 3180 | Median=61 months | Composite recurrence and/ or MSM | Recurrence=496 Dead=277 | Prespecification+univariate analysis | Backward elimination | 3 | Internal-external cross validation |
| Vollmer and Seigler[37] | 1910 | Median=7.6 years | Mortality | NR | Prespecification | Full model approach | 5 | Internal k-fold validation |
| Xiao et al[38] | DEV: 46 336 VALID: 19 856 | 3 and 5 years | OS | NR | Prespecification+univariate analysis | Full model approach | 10 | Internal sample validation |

DEV, development set; DRFS, distant recurrence-free survival; EPV, events per variable; MSM, melanoma specific mortality; MSS, melanoma specific survival; NR, not reported; OS, overall survival; RFS, recurrence-free survival; RRFS, regional recurrence free survival; VALID, validation set.

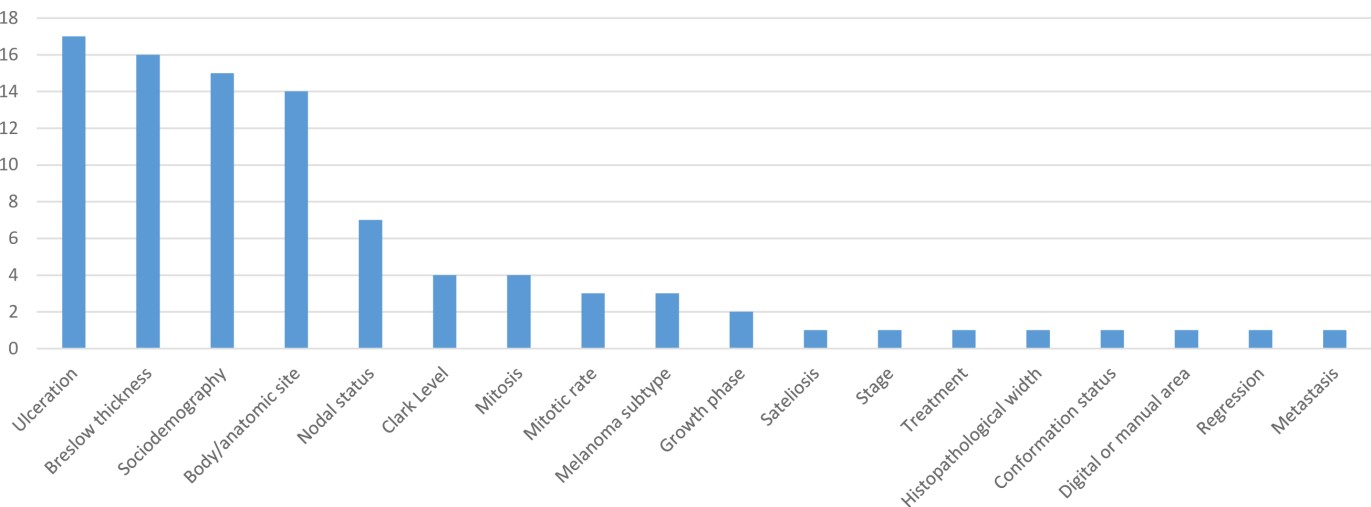

**Figure 1** Identified predictors.

first authors and outcome as follows: Baade MSS model[25]; Balch MSS model[26]; Cochran OS[27]; Cochran recurrence risk score[27]; El Sharouni LRFS nomogram[28]; El Sharouni, RRFS nomogram[28]; El Sharouni DRFS nomogram[28]; Fonseca, DRFS nomogram[29]; Fonseca MSS nomogram[29]; Gimotty metastasis nomogram[31]; Gimotty(a) OS model[30]; Gimotty(b) OS model[30]; Maurichi OS model[32]; Rosenbaum RFS model[33]; Soong OS model[34]; Tsai OS model[35]; Verver MSM model[36]; Vollmer mortality model[37] and Xiao OS model.[38]

### Performance of prediction models

#### Discrimination

The Area Under the Curve (AUC) - Receiver Operating Characteristics (ROC), also known as AUC-ROC was reported in two models (reported in one study) predicting OS,[30] one model predicting RFS[33] and another predicting metastasis outcomes,[31] all with values above 0.7. The C-index measure was reported in the Maurichi OS model[32]; Xiao OS model[38]; El Sharouni DRFS model[28]; Fonseca DRFS model[29]; El Sharouni LRFS model[28]; El Sharouni RRFS model[28]; Fonseca MSS model[29] and Verver MSM model,[36] all with values above 0.7 for the models. The discriminative performance of the models is considered acceptable when the AUC-ROC statistics and their equivalent are ≥0.7.[39] In addition to the C-index, the Baade MSS model also estimated the D-statistic and reported a value of 1.50 (95% CI 1.44 to 1.56) for OS.[25]

#### Calibration

Calibration scores were reported in the Soong OS model,[34] with correlation coefficients of 0.90 and 0.93 for 5-year and 10-year survival rates.[34] Calibration plots for predicting survival outcomes were presented in 10 models: Maurichi OS model[32]; Xiao OS model[38]; El Sharouni DRFS model[28]; Fonseca DRFS model[29]; El Sharouni RRFS model[28]; El Sharouni LRFS model[28]; Fonseca MSS model[29]; Verver recurrence model[36] and Verver MSM model.[36] Three models that reported calibration scores reported values closer to 1, indicating

a perfect agreement, while eight models graphically presented plots showing a good match between predicted and observed outcomes for OS, DRFS, RRFS, LRFS, MSS and metastasis. This suggests that the models have the ability to accurately generate predictions that are close to the observed outcomes. Results indicated high accuracy and precision of the prediction models, as predicted and actual survival probabilities were similar.[40]

#### Overall performance

The Baade OS model assessed the overall performance of predicting OS by assessing how well the model fit the data using the r-squared ($R^2$) statistic.[25] The model reported the $R^2$ as 0.47 (0.45 to 0.49), equating to a strong fit.[25] The Tsai OS model,[35] measured overall performance by assessing the Brier score, with usual values ranging from 0 (total accuracy) to 1 (wholly inaccurate).[41] Results show that the score for the proposed, prognostic classification scheme increased with time from 0.02 at year 1 to approximately 0.20 at year 15, indicating high accuracy in predicting OS.[35]

#### Clinical utility

None of the studies conducted either a net benefit or decision curve analysis, therefore we were unable to address the clinical utility of the tools. The clinical utility of a validated risk prediction model would help clinicians and patients during the surveillance phase of care.[42]

#### Model validation methods

Model validation methods are presented in Online supplemental table 6. Internal validation refers to the performance of a model developed and tested on a sample of the same population.[43] External validation refers to how well a model predicts an outcome in a data set that is different from the development population (new cohort), or a different region or country (geographical).[44] Eight models were internally validated. The Rosenbaum RFS model,[33] and Vollmer mortality model,[37] used the k-fold cross-validation procedure, where an original sample is

randomly partitioned into k equal-sized subsamples.[45] The Cochran OS model,[27] Cochran recurrence model[27] and Xiao OS model,[38] randomly selected a proportion of the sample and retained these as the validation set, using the rest for training. The Maurichi OS model[32] was validated by assessing the congruence of expected outcomes (predicted from the model) and observed outcomes. The Fonseca DRFS model,[29] and Fonseca MSS model,[29] were internally validated using bootstrap methods. Bootstrapping entails repeatedly drawing random samples with replacement from the data to construct prediction models and evaluating model performance using out-of-bag observations.[41] Five models validated their models using external techniques, including geographic validation: the Gimotty(a) OS model[30]; Gimotty(b) OS model[30]; Balch MSS model[26]; Soong OS model[34] and Gimotty metastasis model.[31] Seven models: El Sharouni LRFS model[28]; El Sharouni RRFS model[28]; El Sharouni DRFS model[28]; Verver recurrence model[36]; Verver MSM model[36]; Baade MSS model[25] and the Tsai OS model,[35] assessed model performance using internal-external cross-validation methods (IECV). IECV analyses the performance of models using new patients from different but related practices as compared with the original development sample.[41]

### Critical appraisal of included studies

The results of the critical appraisal are presented in Online supplemental table 7. Overall, eight models: Cochran OS model[27]; Cochran recurrence model[27]; Fonseca DRFS model[29]; Fonseca MSS model[29]; Verver recurrence model[36]; Verver MSM model[36]; Rosenbaum RFS model[33] and the Xiao OS model,[38] were judged to be at high risk of bias. Three models: Maurichi OS model[32]; Soong OS model[34] and Xiao OS model,[38] were rated as low risk. Nine models: El Sharouni LRFS model[28]; El Sharouni RRFS model[28]; El Sharouni DRFS model[28]; Gimotty(a) OS model[30]; Gimotty(b) model[30]; Baade OS model[25]; Balch OS model[26]; Gimotty metastasis model[30] and Vollmer mortality model.[37] The most notable cause of high risk was for the analysis domain. This was mainly due to the inclusion of variables in the final model, previously identified as significant following a univariate analysis.[27 29 33 38] This method can lead to incorrect predictor selection because predictors are chosen on the basis of their statistical significance as a single predictor rather than in context with other predictors.[41]

The applicability ratings of the models are presented in Online supplemental table 8. Two models: Cochran OS model[27] and Cochran recurrence model, were rated as having a high risk regarding applicability, the rest of the models were rated as low risk of bias regarding applicability. Detailed information on critical appraisal is available in Online supplemental file 1. The main concern regarding applicability was for the selection of participants in the study by Cochran et al,[27] where little information is reported about the characteristics of the patients or the severity of their melanoma.

## DISCUSSION

This review identified 14 studies describing 20 different models developed for the prediction of recurrence, new primary tumours or metastasis in patients with AJCC stage I or II cutaneous melanoma following excision. The models differed in the predictors used depending on the outcome of interest and statistical measures used to assess model performance. It was therefore inappropriate to statistically synthesise their results. One of the limitations of the studies was the lack of reporting of baseline rates of SLNB, a technique useful for providing disease stage and to guide adjuvant systemic therapy.[46] The absence of these rates makes it difficult to be sure that the patients are correctly classified in the studies. As the AJCC classification criteria have changed over time, studies from different periods used different staging systems and methodologies. For models at stages I or II we considered which AJCC staging criteria would apply taking into account reported factors such as tumour thickness, and ulceration (see table 1). However, this can lead to significant heterogeneity in the data, making it difficult to compare and synthesise findings. Common risk factors included in the models were consistent with well-established risk factors for melanoma, including ulceration, age, sex, Breslow thickness and tumour site.[47] Although systematic reviews of prediction models for cutaneous melanoma[11 12] have previously been conducted, to the best of our knowledge, this review investigates the potential of clinical models to predict recurrence, metastases and survival of AJCC stage I and II melanoma following surgical excision. Similar to the reviews by Vuong et al,[12] and Mahar et al,[14] notable differences in the approaches used to select predictors during model development were observed. Eight models selected predictors based on univariate analysis of the strength of these predictors, of which four were developed based on previous knowledge. When developing models, building on previous literature and expert clinician opinions is recommended. Building models based only on the statistical significance of the association between predictor and outcome in univariate analyses risks missing important predictors.[48]

Model performance measures were available for assessing discrimination in 14 models, callibration in 11 models and overall performance in 2 models. While the type of discrimination statistic varied, the discrimination statistics of the new models ranged from 0.72 to 0.88, a range comparable to that reported in other published melanoma reviews: 0.62 to 0.86,[12] 0.7 to 0.8[11] and 0.62 to 0.98.[14] Currently, there is a lack of evidence for the clinical performance of early stage melanoma prediction models based on clinicopathological variables. Accuracy measures such as calibration and discrimination, however, do not provide any detail regarding a models suitability for clinical practice. Net benefit, obtained via decision curve analysis, aims to assess the clinical utility of such models.[42] Unfortunately, the studies included in this review do not give detail on clinical utility for the models they report on. In addition, we reviewed follow-up schedules as recommended by clinical guidelines and note the limited evidence

base on which they are based.[49] This suggests more detailed studies about timing and frequency of relapse are needed. Most of the models were externally validated, either in new cohorts or patients from a different location or through IECV. External validation of models is essential to support the general applicability of any prediction model.[48] However, most of the models were rated as being at high risk of bias implying low confidence in the performance of these models in new datasets. This is because a high risk of bias can lead to either overestimation or underestimation of the predictive accuracy of a model, which can affect its generalisability to new datasets.[44]

This review followed procedures documented by the Cochrane Collaboration for conducting systematic reviews, the CHARMS[20] guidelines for extracting data, the PRISMA[16] guidelines for reporting and PROBAST tool[21] for assessing risk of bias, so was therefore robust. The update search conducted in September 2021 incorporated emerging trial evidence, so we are confident that we would have identified relevant trials/studies coming to light in the next 1–2 years from the search implementation. Although the PROBAST tool provides a structured framework for evaluating various aspects of study methodology, like any other tool, it has its limitations. From our limited experience of the tool, we found that its effectiveness relied heavily on the accuracy and comprehensiveness of information available in the included studies. Most of the studies incorporated within our analysis were conducted before guidelines for conducting prognostic studies were developed, leading to missing or inconsistent information, thereby impeding a comprehensive application of the PROBAST tool to adequately appraise bias. A critical analysis of the PROBAST tool was critical of the inter-rater reliability (IRR) of the tool before training.[50] Although we did not calculate the IRR, we assessed RoB in two pairs to ensure consistent and reliable assessments.

A recent guideline has been developed in 2015, the Transparent Reporting of a multivariable prediction model for Individual Prognosis or Diagnosis (TRIPOD), to ensure comprehensive reporting of critical methodological elements in individual prognostic studies.[51] Nevertheless, it is imperative to acknowledge that a limitation of the current review stems from the fact that the included studies were published prior to the availability of this reporting guidance. Recent evaluation of study quality based on the TRIPOD statement reveals insufficient reporting levels for risk prediction models in cutaneous melanoma. Embracing reporting guidelines such as TRIPOD, can enhance future research reporting standards.[52] Second, none of the studies reported a decision curve analysis to assess the clinical implications of the models. Third, comparing the included models was problematic and meta-analysis was not feasible due to the variety in the predictors and statistical measures used for model performances. Finally, we were unable to access the development and validation studies of some of the publicly available online tools, and these were excluded from our analysis.[53 54]

The results of this systematic review highlight the relative lack of appropriate evidence underpinning current melanoma prediction tools to support practice in AJCC stage I and II disease. The evidence needed for clinical guideline decision-makers to incorporate a prediction model into routine clinical care needs to be far more convincing than what has been reported thus far. The goal should be clear demonstration of external validity, by evaluating model transportability in other cohorts,[10] and clinical utility (the ability to better select patients from high-risk groups for adjuvant therapy in the future) by undertaking decision curve analysis to identify net benefits.[42 55] Additionally, biomarkers play a crucial role in risk prediction models of cutaneous melanoma.[56] These may include genetic, immune or circulatory biomarkers.[56] They are measurable characteristics that provide information about the disease's presence, progression and response to treatment.[56] The relationship between biomarkers and clinicopathological characteristics has been shown to be interconnected in predicting chemotherapy response in breast cancer.[57] Therefore, incorporating biomarkers into risk prediction models alongside clinicopathological factors, may provide additional information to refine risk prediction models in cutaneous melanoma and aid healthcare professionals obtain more comprehensive and accurate assessments of an individual's risk. Additionally, other studies have shown that mutation status is associated with survival outcomes,[58 59] however we did not evaluate these studies as they were not developed into risk models. Critically, these results clearly outline the need for ongoing prognostication studies and, as such, this review acts as an evidence base to catalyse project development and funding, and future follow-up guidelines and management of patients, given the relative scarcity of evidence-based practice at present.

## CONCLUSION

The data, which are heterogeneous in terms of biology and progression, do not offer a wide enough scope of best practice to allow accurate prognostication of melanoma AJCC stages I or II patients as defined by the criteria within the third, sixth, seventh and eighth editions of AJCC staging, or recommendation for use in clinical practice. This raises concern as to whether current follow-up recommendations postsurgical treatment is adequate as the evidence supporting such recommendation is sparse. Future research should focus on validating existing models utilising TRIPOD guidelines to improve reporting quality. Future studies should also look to use decision curve analysis to analyse the net benefit of using the predictive model.[55 60]

**Author affiliations**
[1]Evidence Synthesis Group, Population Health Sciences Institute, Newcastle University, Newcastle upon Tyne, UK
[2]NIHR Innovation Observatory, Population Health Sciences Institute, Newcastle University, Newcastle upon Tyne, UK
[3]Biostatistics Research Group, Population Health Sciences Institute, Newcastle University, Newcastle upon Tyne, UK
[4]Health Economics Group, Population Health Sciences Institute, Newcastle University, Newcastle upon Tyne, UK

[5]Dermatological Sciences, Translation and Clinical Research Institute, Newcastle University, Newcastle upon Tyne, UK
[6]AMLo Biosciences, The Biosphere, Newcastle Helix, Newcastle upon Tyne, UK
[7]Department of Dermatology, South Tees Hospitals NHS FT, Middlesbrough, UK

**Acknowledgements** We would like to thank our colleagues from the Evidence Synthesis Group, and NIHR Innovation Observatory, of Newcastle University and Dr Batoul Nasr from Dermatological Sciences, Translational Research Institute, Newcastle University for their support in screening.

**Collaborators** N/A.

**Contributors** TPK was responsible for the conceptualisation of this project, methodology, writing of the original draft, visualisation of results, the overall project administration and is the guarantor of this work. RE codesigned the protocol, undertook all stages of the review and cowrote the paper. RPWK, MA, VK and DCoughlan undertook all stages of the review and cowrote the paper. AB codesigned the protocol undertook all stages of the review and cowrote the paper. CR and CHE undertook the searches for the review and cowrote the paper. FB, DCraig, PI and LV codesigned the protocol and cowrote the paper. FP contributed to the analysis of review data and cowrote the paper.

**Funding** This work was supported by the National Institute for Health Research (NIHR) Health Technology Assessment Programme, grant number 16/166/05 and the NIHR Invention for Innovation (i4i) Innovative Prognostic Test for Early-Stage Cutaneous Melanoma, grant number 20993.

**Competing interests** TPK, RPWK, MA, AB, VK, DCoughlan, CR, CHE, FB and FP have no conflicts of interest to declare. DCraig: December 2018 to present, member of the HD&DR Research-led prioritisation committee. LV was a member of the NIHR Clinical Evaluation and Trial Panel from 2015 to 2018. LV and DCraig are both part funded by the NIHR Applied Research Collaboration for the North East and North Cumbria. PL is Chief Scientific Officer for AMLo Biosciences, holds shares in AMLo and is named as an inventor on patents for biomarkers this area. RE received personal fees from AMLo Biosciences, outside the submitted work, he also was previously Chief Medical Officer for AMLo Biosciences but gave up this position in January 2021. He is named as an inventor on patents owned by AMLo Biosciences for biomarkers this area.

**Patient and public involvement** Patients and/or the public were not involved in the design, or conduct, or reporting, or dissemination plans of this research.

**Patient consent for publication** Not applicable.

**Ethics approval** Not applicable.

**Provenance and peer review** Not commissioned; externally peer reviewed.

**Data availability statement** All data relevant to the study are included in the article or uploaded as supplementary information.

**ORCID iD**
Tafadzwa Patience Kunonga http://orcid.org/0000-0002-6193-1365

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
