## [Reviewer comments · BMJ Open]

ARTICLE DETAILS

TITLE (PROVISIONAL)	Predictive Accuracy of Risk Prediction Models for Recurrence, Metastasis and Survival for Early-Stage Cutaneous Melanoma: A Systematic Review
AUTHORS	Kunonga, Tafadzwa; Kenny, RPW; Astin, Margaret; Bryant, Andrew; Kontogiannis, Vasileios; Coughlan, Diarmuid; Richmond, Catherine; Eastaugh, Claire H.; Beyer, Fiona; Pearson, F; Craig, Dawn; Lovat, Penny; Vale, Luke; Ellis, Robert

VERSION 1 – REVIEW

REVIEWER	Vuong, Kylie Griffith University
REVIEW RETURNED	19-Jun-2023

GENERAL COMMENTS	Thank you for the opportunity to be part of the peer review process. This study aims to review risk prediction models for recurrence, metastasis and survival for early-stage cutaneous melanoma. Some comments below: 1. Page 8, Methods, line 56 Introduce the constructs of the PROBAST tool in more detail, including how were these were interpreted and assessed by the reviewers2. Page 10, Results, through out Consider naming the models to enable them to be clearly identified in text3. Page 11, Results, Clinical utility It would be worth addressing impact to link in with the introduction.4. Page 11, Results, Model validation methods As above, consider naming the models to enable them to be clearly identified in text.5. Page 12, Results, Critical appraisal Provide reasons for how the reviewers arrived at their assessment of bias6. Page 14, Discussion, line 17 Please provide more details on the assessment of bias and how it is associated with "low confidence in the performance of these models in new datasets"7. Page 14, Discussion, line 32-36 This sentence appears to be a repetition of the previous one.8. Page 14, Discussion line 37-40 May be more appropriate in the Methods section.9. Page 14, Discussion, line 47-54 May be more appropriate in the Results section. These appears to be a limitation of the individual models, not the review.10. Page 15, Discussion, line 22-31 The role of biomarkers in risk prediction and its relationship with the study results could be made clearer.
--

	11. Page 15, Conclusion The TRIPOD guideline is a recent document. By focusing only on models that utilising the TRIPOD guidelines, older studies as well as those which don't report accordingly will be excluded. It's also not clear how the authors have arrived at this conclusion. 12. Table 2 Consider including more information about the data source, for example study design and sample size, and including details on the development and validation methods. Overall, a well-written and comprehensive review. The paper could benefit from stronger alignment of the individual sections. It would be worthwhile to include some suggestions on which model/s (if any) might be most suitable for clinical use based on current evidence and the underpinning rationale.
--	--

REVIEWER	Pfahlberg, Annette Friedrich-Alexander-Universität Erlangen-Nürnberg, Medical Informatics, Biometry and Epidemiology
REVIEW RETURNED	25-Jun-2023

GENERAL COMMENTS	The manuscript reports results from a systematic review (SR) on clinical risk prediction models for survival, recurrence, and metastasis of early-stage melanoma. The SR has been well planned, properly conducted, and comprehensively reported following the PRISMA guideline. It addresses a topic that has not been analysed before (at least in this specific form) and thus constitutes a valuable and interesting contribution. Specific remarks:  - In the Introduction three earlier SRs are mentioned. Two of them (Vuong et al. and Usher-Smith et al.) described risk prediction models for melanoma development, only the SR by Mahar et al. is devoted to clinical risk prediction models for survival of melanoma cases. This should be clearly distinguished. Regarding melanoma development a more recent SR is available that should be incorporated (https://doi.org/10.3390/ijerph17217919). - In the Discussion the TRIPOD guideline for reporting prediction models is mentioned (p. 12). Without giving details the authors state that "none of the studies reported having followed the Transparent Reporting of a multivariable prediction model for Individual Prognosis or Diagnosis (TRIPOD) or met the criteria outlined". It would be interesting to add some more details in the Results section (or as part of the supplement) to corroborate this statement. Adherence to TRIPOD is not a dichotomous criterion, it is typically measured using an ordinal score. Two recent studies addressed TRIPOD adherence among melanoma prediction studies (see https://doi.org/10.1016/j.surg.2020.04.016 and https://doi.org/10.3390/healthcare10020238) and described disappointing results that would match the authors' claim. - The SR uses PROBAST to assess the risk of bias (RoB) of the prediction models. According to a meta-review by de Jong et al.
---

	(https://doi.org/10.1111/nep.13913) PROBAST has gained in popularity very fast. It has already been used for melanoma prediction studies before which should be mentioned (see https://doi.org/10.3390/cancers14123033). One peculiar aspect of PROBAST is its low inter-rater agreement that has recently been criticized (see https://doi.org/10.3390/jcm12051976). The authors should give more details on their PROBAST experience in this SR. Have inter-rater agreement measures been calculated (for the domain-specific RoB and the overall RoB)? The aspect of (low) study quality is presented as the primary result of this SR. Some more detail and reflection on the RoB tool used for measuring study quality is thus highly welcome. A very recent meta-review to be published in July (https://doi.org/10.1016/j.jclinepi.2023.04.012) takes the same line of criticism as the earlier meta-review. - The criteria of the American Joint Committee on Cancer (AJCC) for classifying early-stage melanoma into stage I and II have been revised more than once over the last decades. Thus, the meaning of stage I/II melanoma in the early studies entering the SR might be different from the more recent studies. How have the authors handled this aspect? The manuscript touches upon the issue of the different AJCC editions, but I did not get how it was dealt with in the SR (it is not mentioned in the limitations section, thus I assume that the authors have found a way to handle the heterogeneity). - I missed a rationale (other than the statement on p. 13 "beyond the scope of this review") for the decision to exclude all prediction models incorporating predictive biomarker factors and/or gene expression profiling from the SR. To my opinion, it is inconsistent to intentionally exclude this large group of studies from the SR and simultaneously state in the abstract that "future models should consider incorporating biomarker factors as they may increase the predictive accuracy of these models". According to Supplemental Figure 1 the authors have found 29 publications reporting on prediction models including biomarkers. Thus, the authors could have answered the question whether the inclusion of these predictors increases the predictive accuracy of the models or not. - The very short description of bootstrapping on p. 10 has to be rephrased as it is misleading in this abbreviated form. The main idea of bootstrapping as a method for the internal validation of a prediction model is not captured in the short description. Bootstrapping involves the drawing of random samples with replacement repeatedly from the data to build the prediction models and using the out-of-bag observations for the repeated evaluation of the model performance. - In Table 3 the abbreviation MSM (melanoma specific mortality) is used, but not explained in legend.
--	---

VERSION 1 – AUTHOR RESPONSE

Reviewer 1

1. Page 8, Methods, line 56

Introduce the constructs of the PROBAST tool in more detail, including how were these were interpreted and assessed by the reviewers

Our response

Thank you very much for comment. We have added the following paragraph to the Risk of bias section on page 7, to give more detail to the constructs to the PROBAST tool:

“Two pairs of reviewers using the Prediction study Risk of Bias Assessment Tool (PROBAST) assessed each study for risk of bias (ROB). The tool evaluates 20 questions in four domains (Participants, Predictors, Outcome, and Analysis) to assess the ROB and applicability of prediction model studies. We used the signalling questions across these domains to make the following judgements: “yes” means low bias, “no” means high bias. For the overall judgement, the study was rated as low ROB if all domains were rated as low, high ROB if at least one domain was rated as high. Where there is insufficient information to make a judgement, we rated the domain as unclear ROB. If one domain was rated as unclear and the others low, overall judgement was rated as unclear.”

2. Page 10, Results, through out. Consider naming the models to enable them to be clearly identified in text.

Our response

Thank you very much for your comment. Throughout the results section, we have named the models based on first author and outcome predicted to distinguish between some multiple models developed in one study.

The models are reported based on first authors and outcome as follows: Baade MSS model, Balch MSS model, Cochran OS risk score, Cochran recurrence risk score, El Sharouni LRFS nomogram, El Sharouni RRFS nomogram, El Sharouni DRFS nomogram, Fonseca, DRFS nomogram; Fonseca MSS nomogram, Gimotty metastasis nomogram, Gimotty (a) OS model, Gimotty(b) OS model, Maurichi OS model, Rosenbaum RFS model, Soong OS model, Tsai OS model, Verver MSM model, Vollmer mortality model, and Xia OS model.

3. Page 11, Results, Clinical utility: It would be worth addressing impact to link in with the introduction.

Our response

Thank you very much for your comment. We have added the following statement on page 10 in the results section under clinical utility:

“The clinical utility of a validated risk prediction model would help clinicians and patients during the surveillance phase of care.”

4. Page 11, Results, Model validation methods

As above, consider naming the models to enable them to be clearly identified in text.

Our response

Please see response to (2) above.

5. Page 12, Results, Critical appraisal

Provide reasons for how the reviewers arrived at their assessment of bias

Our response

Please see Supplementary File 1: Risk of bias assessment in the Supplementary Materials on pages 15 to 16, for further details on how we arrived at our assessments.

6. Page 14, Discussion, line 17

Please provide more details on the assessment of bias and how it is associated with “low confidence in the performance of these models in new datasets”

Our response

Thank you very much for your comment.

We have added the following statement below, at the end of the first paragraph on page 13 to show how bias is associated with low confidence in a new dataset:

“This is because a high risk of bias can lead to either overestimation or underestimation of the predictive accuracy of a model, which can affect its generalisability to new datasets.”

7. Page 14, Discussion, line 32-36

This sentence appears to be a repetition of the previous one.

Our response

Thank you very much for your comment. We have deleted the following statements below on page 13 as it was a repetition of the one before:

“Having updated this review previously we can also confidently say that the trajectory of research relevant to the review research question is very low. Given this, we are confident that the findings of

the review would not change substantively following an update.”

8. Page 14, Discussion line 37-40

May be more appropriate in the Methods section.

Our response

Thank you very much for your comment, we have moved the discussion of the GRADE assessment to the methods section from page 13 to page 7 under “Quality of the evidence” section.

9. Page 14, Discussion, line 47-54

May be more appropriate in the Results section. These appears to be a limitation of the individual models, not the review.

Our response

Thank you very much for your comment. We have now moved the paragraph to the results section on pages 9 and 10 under the Discrimination and Calibration sections.

10. Page 15, Discussion, line 22-31

The role of biomarkers in risk prediction and its relationship with the study results could be made clearer.

Thank you very much for your comment. We have revised this section to make it a bit more clearer and added the following to the third paragraph on page 14:

“Additionally, biomarkers play a crucial role in risk prediction models of cutaneous melanoma. These may include genetic, immune or circulatory biomarkers. They are measurable characteristics that provide information about the disease’s presence, progression, and response to treatment. The relationship between biomarkers and clinico-pathological characteristics has been shown to be interconnected in predicting chemotherapy response in breast cancer. Therefore, incorporating biomarkers into risk prediction models alongside clinico-pathological factors, may provide additional information to refine risk prediction models in cutaneous melanoma and aid healthcare professionals obtain more comprehensive and accurate assessments of an individual’s risk.”

11. Page 15, Conclusion

The TRIPOD guideline is a recent document. By focusing only on models that utilising the TRIPOD guidelines, older studies as well as those which don’t report accordingly will be excluded. It’s also not clear how the authors have arrived at this conclusion.

Our response

Thank you very much for your comment. We refer to the TRIPOD guideline as a recommendation for use in future individual prognostic studies to ensure key methodological aspects are reported. We hope that adhering to the guideline will enhance study credibility, reproducibility, and comparability, ultimately improving the quality of evidence for clinical decision-making and promoting advancement in prognostic research.

12. Table 2

Consider including more information about the data source, for example study design and sample size, and including details on the development and validation methods.

Our response

Thank you very much for your comment. The details regarding study design and data source, development method are provided in Table 2. The details regarding sample size are provided in Table 3 for model development. We have added a column on validation method in Table 3. Please also refer to further discussions on validation methods within the text under; “Model validation methods” on pages 10-11.

13. Overall, a well-written and comprehensive review. The paper could benefit from stronger alignment of the individual sections. It would be worthwhile to include some suggestions on which model/s (if any) might be most suitable for clinical use based on current evidence and the underpinning rationale.

Our response

Thank you very much for your comment. We have improved the alignment of the manuscript accordingly. Please see responses to query (8) and (9) above.

Unfortunately, we were unable to assess clinical performance of the models because the models do not report any detail on clinical utility, therefore we could not make a judgement on which of these models would be the most suitable based on the evidence at hand. We made reference to this in our discussion in the first paragraph on page 13 where we stated:

“Accuracy measures such as calibration and discrimination, however, do not provide any detail regarding a models suitability for clinical practice.”

Reviewer 2

1. The manuscript reports results from a systematic review (SR) on clinical risk prediction models for survival, recurrence, and metastasis of early-stage melanoma. The SR has been well planned,

properly conducted, and comprehensively reported following the PRISMA guideline. It addresses a topic that has not been analysed before (at least in this specific form) and thus constitutes a valuable and interesting contribution.

Our response

Thank you very much for your encouraging comment.

2. In the Introduction three earlier SRs are mentioned. Two of them (Vuong et al. and Usher-Smith et al.) described risk prediction models for melanoma development, only the SR by Mahar et al. is devoted to clinical risk prediction models for survival of melanoma cases. This should be clearly distinguished. Regarding melanoma development a more recent SR is available that should be incorporated (<https://doi.org/10.3390/ijerph17217919>)

Our response

Thank you very much for your comment. We have rephrased the sentence in the last paragraph on page 5 to the following:

“Previous systematic reviews have been conducted looking at the value of risk prediction models for melanoma development (Vuong et al; Kaiser et al; and Usher Smith et al) and clinical risk of survival of melanoma (Mahar et al).”

3. In the Discussion the TRIPOD guideline for reporting prediction models is mentioned (p. 12). Without giving details the authors state that “none of the studies reported having followed the Transparent Reporting of a multivariable prediction model for Individual Prognosis or Diagnosis (TRIPOD) or met the criteria outlined”. It would be interesting to add some more details in the Results section (or as part of the supplement) to corroborate this statement. Adherence to TRIPOD is not a dichotomous criterion, it is typically measured using an ordinal score. Two recent studies addressed TRIPOD adherence among melanoma prediction studies (see <https://doi.org/10.1016/j.surg.2020.04.016> and <https://doi.org/10.3390/healthcare10020238>) and described disappointing results that would match the authors' claim.

Our response

Thank you very much for your comment, we have rephrased limitations of the review based on non-adherence to TRIPOD in the second paragraph on page 13 to the following:

“A recent guideline has been developed in 2015, the Transparent Reporting of a multivariable prediction model for Individual Prognosis or Diagnosis (TRIPOD) to ensure comprehensive reporting of critical methodological elements in individual prognostic studies. Nevertheless, it is imperative to acknowledge that a limitation of the current review stems from the fact that the included studies were published prior to the availability of this reporting guidance. Recent evaluation of study quality based on the TRIPOD statement reveals insufficient reporting levels for risk prediction models in cutaneous melanoma. Embracing reporting guidelines such as TRIPOD, can enhance future research reporting standards.”

4. The SR uses PROBAST to assess the risk of bias (RoB) of the prediction models. According to a meta-review by de Jong et al. (<https://doi.org/10.1111/nep.13913>) PROBAST has gained in popularity very fast. It has already been used for melanoma prediction studies before which should be mentioned (see <https://doi.org/10.3390/cancers14123033>). One peculiar aspect of PROBAST is its low inter-rater agreement that has recently been criticized (see <https://doi.org/10.3390/jcm12051976>). The authors should give more details on their PROBAST experience in this SR. Have inter-rater agreement measures been calculated (for the domain-specific RoB and the overall RoB)? The aspect of (low) study quality is presented as the primary result of this SR. Some more detail and reflection on the RoB tool used for measuring study quality is thus highly welcome. A very recent meta-review to be published in July (<https://doi.org/10.1016/j.jclinepi.2023.04.012>) takes the same line of criticism as the earlier meta-review.

Our response

Thank you very much for comment. We have added the de Jong reference to the risk of bias section on page 7. We have also added the following statement in the discussion in the last paragraph on page 14 -15 regarding our PROBAST experience:

Although the PROBAST tool provides a structured framework for evaluating various aspects of study methodology, like any other tool, it has its limitations. From our experience, we found that its effectiveness relied heavily on the accuracy and comprehensiveness of information available in the included studies. Most of the studies incorporated within our analysis were conducted before guidelines for conducting prognostic studies were developed, leading to missing or inconsistent information, thereby impeding a comprehensive application of the PROBAST tool to adequately appraise bias. A critical analysis of the PROBAST tool was critical of the inter-rater reliability (IRR) of the tool before training. Although we did not calculate the IRR, we assessed RoB in two pairs to ensure consistent and reliable assessments.

5. The criteria of the American Joint Committee on Cancer (AJCC) for classifying early-stage melanoma into stage I and II have been revised more than once over the last decades. Thus, the meaning of stage I/II melanoma in the early studies entering the SR might be different from the more recent studies. How have the authors handled this aspect? The manuscript touches upon the issue of the different AJCC editions, but I did not get how it was dealt with in the SR (it is not mentioned in the limitations section, thus I assume that the authors have found a way to handle the heterogeneity).

Our response

Thank you very much for your comment. We were aware of these changes at the commencement of this project. For models at stages I or II we, considered which AJCC staging criteria would apply (e.g, ulceration, thickness), and we referred to this in Table 1.

We also added the following statement in the discussion section on page 12:

“As the AJCC classification criteria have changed over time, studies from different periods used different staging systems and methodologies. For models at stages I or II we, considered which AJCC staging criteria would apply taking into account reported factors such as tumour thickness, and ulceration (see Table 1). However, this can lead to significant heterogeneity in the data, making it difficult to compare and synthesize findings.”

6. I missed a rationale (other than the statement on p. 13 "beyond the scope of this review") for the decision to exclude all prediction models incorporating predictive biomarker factors and/or gene expression profiling from the SR. To my opinion, it is inconsistent to intentionally exclude this large group of studies from the SR and simultaneously state in the abstract that "future models should consider incorporating biomarker factors as they may increase the predictive accuracy of these models". According to Supplemental Figure 1 the authors have found 29 publications reporting on prediction models including biomarkers. Thus, the authors could have answered the question whether the inclusion of these predictors increases the predictive accuracy of the models or not.

Our response

Thank you very much for your comment. The focus of this review was on patient characteristics in order to provide a baseline of their physical characteristics, and clinical factors of the melanoma for comparison with future developments, e.g., age, Breslow depth, ulceration, disease stage, skin type, lymph node involvement. We acknowledge the synthesised evidence on biomarkers, including gene expression profiling (GEP) to understand if they increase the accuracy of prediction of metastatic risk. We are now advocating for the synthesis of clinic-pathological as well as biomarkers to assess whether they may increase predictive accuracy. Please see response to query 10 from Reviewer 1 above.

7. The very short description of bootstrapping on p. 10 has to be rephrased as it is misleading in this abbreviated form. The main idea of bootstrapping as a method for the internal validation of a prediction model is not captured in the short description. Bootstrapping involves the drawing of random samples with replacement repeatedly from the data to build the prediction models and using the out-of-bag observations for the repeated evaluation of the model performance.

Our response

Thank you very much for your comment, we have rephrased the description of bootstrapping on page 11 to the following:

“Bootstrapping entails repeatedly drawing random samples with replacement from the data to construct prediction models and evaluating model performance using out-of-bag observations.”

8. In Table 3 the abbreviation MSM (melanoma specific mortality) is used, but not explained in legend.

Our response

Thank you very much for your comment. We have added MSM = Melanoma Specific Mortality in the legend to Table 3.